# Classification for Breast Ultrasound Using Convolutional Neural Network with Multiple Time-Domain Feature Maps

**Hyungsuk Kim** (ID)**, Juyoung Park, Hakjoon Lee, Geuntae Im, Jongsoo Lee, Ki-Baek Lee *** (ID) **and Heung Jae Lee *** (ID)

Department of Electrical Engineering, Kwangwoon University, Seoul 01897, Korea; hskim@kw.ac.kr (H.K.); 419kog@kw.ac.kr (J.P.); cpfl410@kw.ac.kr (H.L.); holsky7@kw.ac.kr (G.I.); dldnxks12@kw.ac.kr (J.L.)
* Correspondence: kblee@kw.ac.kr (K.-B.L.); hjlee@kw.ac.kr (H.J.L.)

**Abstract:** Ultrasound (US) imaging is widely utilized as a diagnostic screening method, and deep learning has recently drawn attention for the analysis of US images for the pathological status of tissues. While low image quality and poor reproducibility are the common obstacles in US analysis, the small size of the dataset is a new limitation for deep learning due to lack of generalization. In this work, a convolutional neural network (CNN) using multiple feature maps, such as entropy and phase images, as well as a B-mode image, was proposed to classify breast US images. Although B-mode images contain both anatomical and textual information, traditional CNNs experience difficulties in abstracting features automatically, especially with small datasets. For the proposed CNN framework, two distinct feature maps were obtained from a B-mode image and utilized as new inputs for training the CNN. These feature maps can also be made from the evaluation data and applied to the CNN separately for the final classification decision. The experimental results with 780 breast US images in three categories of benign, malignant, and normal, showed that the proposed CNN framework using multiple feature maps exhibited better performances than the traditional CNN with B-mode only for most deep network models.

**Keywords:** medical ultrasound; breast US images; deep learning; convolutional neural network; B-mode image; entropy image; phase image

## 1. Introduction

Among medical imaging modalities, ultrasound (US) is one of the most commonly utilized in clinical screening and diagnostic applications due to its safety by utilization of non-ionizing radiation, portability, cost effectiveness, and real-time data acquisition and display. Despite these advantages, US imaging also has limitations, such as relatively low imaging contrast and degradation of quality caused by noise and speckles, high image variability due to the operator-dependent hand-held nature in the data acquisition process, and poor image reproducibility across different manufacturers' US imaging systems. Consequently, a more objective and accurate understanding for analysis of US images, called B-mode images, is important for US diagnosis and assessment in addition to ultrasound-guided interventions and therapy.

For the better analysis of US images, computer-aided diagnosis (CAD) systems using machine learning algorithms have been developed and applied to various kinds of features that are calculated and/or estimated from the B-mode images in order to classify or quantify the pathological status of the scanned tissue. In traditional CAD systems, image features including texture, contrast, pattern, morphology, and model-based parameters are extracted first from B-mode images automatically or manually and then selected and classified using an automatic classifier such as a support vector machine (SVM) [1] to divide the feature space. Since AlexNet [2], which is a convolutional neural network (CNN) in early deep learning generation, won the first prize in the 2012 ImageNet Large Scale Visual Recognition Challenge (ILSVRC), deep learning has garnered significant attention

as a promising machine learning technique, especially in the research domains of natural language processing [3], computer vision [4], and various imaging analyses, including medical images [5,6].

From the perspective of image analysis, the deep learning approach directly processes the original data and automatically learns multiple levels of abstraction features from images through supervised or unsupervised methods. It has achieved state-of-the-art performance in image analysis with large datasets such as ImageNet [7] in various general tasks, including classification, segmentation, tracking, and object detection. In the areas of medical US image analysis, recent applications of deep learning have involved traditional diagnosis tasks such as tissue classification, tumor detection and segmentation, biometric measurements, and quality assessment. It has also been extended to the clinical domain of US-guided interventions and therapy. However, the main barrier for deep learning in US image analysis is the lack of a well-organized dataset in terms of size and annotated classes in the training set. According to recent papers, most studies utilized a few hundred images to train deep learning networks, which is much smaller than datasets in natural image applications. In addition, for the uniformity of datasets in US applications, most databases were generally acquired from a single-vendor's US imaging system in a single institution. These limitations of US image diversity may degrade the performance of a specific task using deep learning techniques because of the lack of generalization of features and the overfitting of a given dataset.

Notwithstanding the above constraints, many studies using deep learning in US image analysis have been conducted and exhibit better performance than traditional machine learning approaches. Breast ultrasound (BUS) is an active application area, since breast cancer is the second leading cause of death for women [8], and early detection and diagnosis is crucial to increase the success of treatment and reduce medical expenses. In 2012, the adaptive deconvolutional network (ADN), which is an unsupervised and generative hierarchical deep model, was applied to classify benign and malignant breast tumors and mass lesions [9]. Along with many other studies of deep learning in US images, Lui et al. proposed a supervised deep learning network, called a deep polynomial network (DPN), for the classification of breast ultrasound images and exhibited the highest performance of 92.4% on small US datasets [10]. In addition to B-mode images, shear-wave elastography, which represents tissue characteristics instead of the anatomical structure, has also been utilized for breast cancer diagnosis [11].

Recently, ensemble network approaches have been utilized with multiple CNN models and/or inputs representing different aspects of an image in either time or frequency domains. Tanaka et al. proposed an ensemble network by combining two CNN models (VGG-19 and ResNet-152) with a heat map and concluded that many breast masses were not detected by conventional CNNs as the important regions for correct classification [12]. Moon et al. proposed an ensemble learning architecture using B-mode image, tumor shape image (TSI), which is manually extracted along mass boundary by an expert, and the segmented tumor image represents the tumor region only [13]. The fusion image made by RGB-like concatenation of three images was also used as one of the inputs for an ensemble network and achieved good classification performances due to manually extracted image features. Ensemble transfer learning architecture using elastography as well as B-mode images was proposed and showed benefits of different features rather than morphological information in breast tumor classification [14].

In this paper, we propose a classification framework for breast US images using a CNN with multiple feature images produced in the time domain. Although the virtue of CNNs is a direct abstraction of image features from the raw data by the learning process, it is difficult to learn some features that are not represented (partially or entirely) in the raw image. For example, because B-mode images are generated only by the magnitude of the echoed radiofrequency signals, the reflections from large organ boundaries or tumors represent the anatomical structure relatively well. However, the magnitude of reflected signals from interior tumors or tissue microstructures, which are generally related to the pathological

status of soft tissue, is relatively small due to the nature of diffuse scattering and absorption of the propagating ultrasonic waves, so it is hardly recognized in a B-mode image. In addition, as mentioned above, this self-learning ability of deep neural networks generally degrades as the size of training datasets gets smaller owing to lesser generalization and overfitting problems. Therefore, additional information (or features) that are extracted from the original data at different aspects may increase the performance of the deep learning network. In this work, the entropy and phase images (called feature maps) generated from a B-mode image are provided to the CNN as new inputs. Because the entropy images generally represent the small local texture of a B-mode image and the phase images show an enhanced morphology including organ boundaries and edge structures, the combination of the three feature maps—B-mode, entropy, and phase images—provides additional information to the CNN, especially for the case of a smaller dataset. The other advantage of the proposed framework is that the same feature maps can be calculated from the new data in the evaluation stage, and applied to the trained convolutional deep network as separate inputs to obtain the classification decision. This means that a single evaluation data sample has three classification decisions in our framework, and these results can be used to obtain the final classification result, either weighted sum or voting schemes.

The remainder of this paper is organized as follows. In the next section, a brief explanation of the dataset and its properties are described. In addition, the concepts of the two feature maps used in this work (entropy image and phase image) and detailed methods to generate them from B-mode images are presented. Section 3 provides a description of the proposed deep convolutional neural network and training parameters. The experimental results are presented and discussed in Sections 4 and 5, respectively. Finally, Section 6 presents the conclusions and future work.

## 2. Materials

Ultrasound images are generally referred to as B-mode images or brightness images, which are constructed from the envelope of reflected or backscattered ultrasound radiofrequency waves from a scanned tissue. The received ultrasound signals at the ultrasonic transducer are divided into two categories: the reflected waves at the boundaries that are relatively much larger than the transmitted wavelength, and the backscattered waves from a large number of randomly distributed scatterers within a soft tissue. While the reflected waves are presented as organ or tumor boundaries in a B-mode image that provide anatomical structures of the human body, the backscattered waves generally result in speckle patterns in a B-mode image that are closely related to tissue microstructures. Although a B-mode image utilizes only the magnitude information of the received ultrasound radiofrequency signals, it can be said that a B-mode image contains both anatomical and microstructural information simultaneously in one image. However, in the analysis of B-mode images, experienced experts are needed for classification, in terms of morphological information and textual features that are related to the microstructure in a B-mode image.

When this classification is executed by convolutional deep neural networks, we generally believe that the deep network learns both types of information—morphological and textual features—from a B-mode image through the training process by itself. However, the low image quality, as well as the relatively small size of datasets in medical ultrasound applications, might become obstacles for deep learning approaches. To compensate for these limitations during the training phase in deep learning approaches, distinct feature maps (or images) to represent tissue characteristics explicitly are used to improve the classification performance. In this section, we briefly introduce the breast ultrasound dataset used in this study and present the methods to calculate the two feature maps, entropy and phase images, from this B-mode dataset.

## 2.1. Breast US Image Dataset

Although US imaging is one of the most popular modalities in practical clinical applications, it is difficult to find publicly available US image datasets in the literature. The dataset used in this study was recently released to the public, which consists of breast US images from 600 female patients aged between 25 and 75 years at the Baheya hospital in 2018 [15]. The data were acquired using the LOGIQ E9 ultrasound system and LOGIQ E9 Agile ultrasound system with the ML6-15-D matrix linear transducer, and converted gray scale images were obtained with an average size of $500 \times 500$ in PNG file format. The dataset consists of 780 B-mode images with pathological status and is categorized into three classes: normal, benign, and malignant. The numbers of images for the three classes are presented in Table 1, and the sample US images of each class are shown in Figure 1.

**Table 1.** Three classes of breast US images and the number of images in each class.

| Class | Number of Images |
|---|---|
| Benign | 487 |
| Malignant | 210 |
| Normal | 133 |
| Total | 780 |

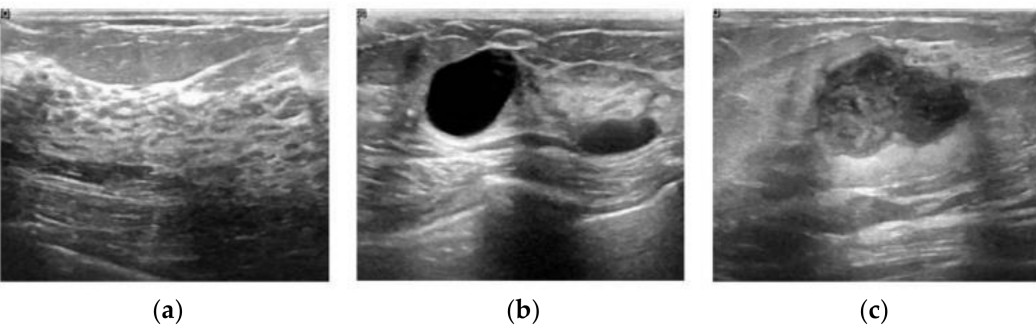

| (a) | (b) | (c) |

**Figure 1.** Sample B-mode images of the breast ultrasounddataset [11]; (**a**) Normal, (**b**) benign, and (**c**) malignant.

## 2.2. Entropy Images

US B-mode images contain the anatomical information formed by reflections at large boundaries as well as microstructural properties inside a small local area, which may be related to the pathological status of soft tissue. These backscattered waves are generally shown as speckles, periodic patterns, or textures in a B-mode image. Therefore, many approaches have been used to analyze tissue characteristics from these randomly backscattered signals using various statistical models [16]. For example, the Nakagami [17], K-distribution [18], and homodyned K-distribution [19] are popular statistical distributions for assessing pathological information. For these approaches, called model-based methods, there are some limitations in estimating the distribution parameters for a specific model because a single model cannot be satisfied for the entire image area, and nonlinear signal processing such as log-compression may change the original statistical characteristics of raw data. Therefore, non-model-based approaches have been studied to compensate for the limitations of statistical model-based methods.

The Shannon entropy, a measure of the average level of information, was proposed in the research area of information theory [20] and applied to many signal processing applications to estimate the signal uncertainty in a random variable. Since Hughes first utilized the Shannon entropy for the analysis of ultrasonic waveforms in a scattering medium [21], many studies using the same concept for ultrasound images have been proposed and exhibited performances comparable to those of model-based approaches [22–24]. The

Shannon entropy of a discrete random variable $X$ with possible values $\{x_1, x_2, \cdots, x_n\}$ is defined in the following discrete form:

$$H = -\sum_{i=1}^{n} p(x_i) \log_2[p(x_i)] \tag{1}$$

where $p(\cdot)$ represents the function of probability distribution.

In this study, the entropy image was constructed from the B-mode image using a small moving rectangular window in both the axial and lateral directions. Although Wan et al. showed that the appropriate window length in the axial direction for a stable statistical parameter was three times the pulse length [25], the window used in this study was a fixed size of $25 \times 25$ pixels, which is the minimum size considering the average size of B-mode images, because detailed information on raw radiofrequency data was not available. The sample entropy images calculated from the B-mode image are shown in Figure 2b.

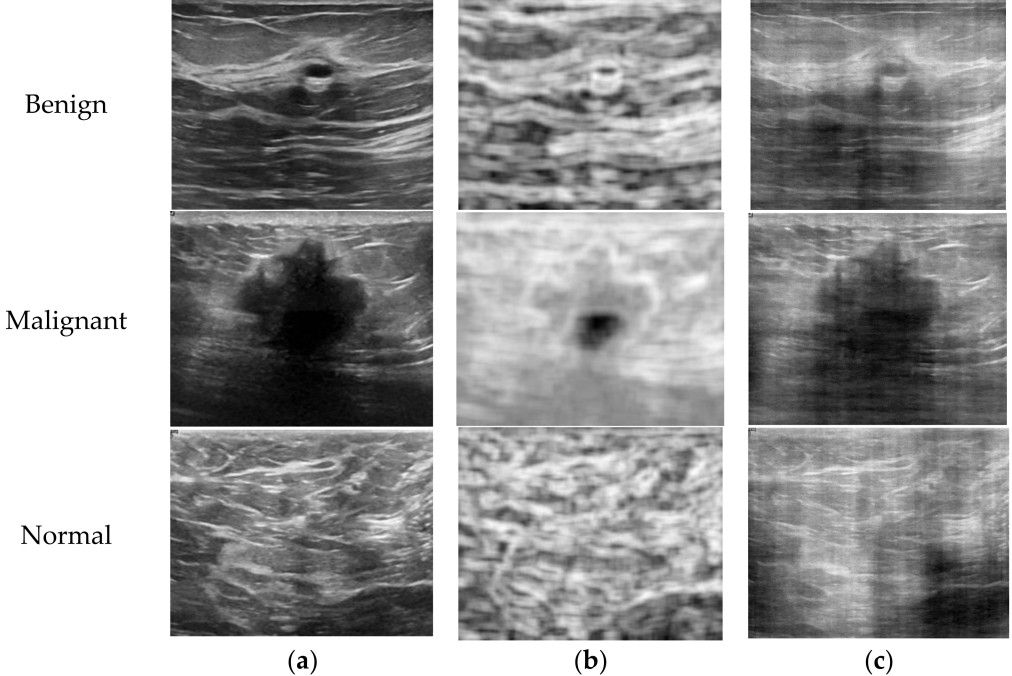

**Figure 2.** Sample B-mode, entropy, and phase images for a patient in each class; (**a**) B-mode images, (**b**) entropy images, and (**c**) phase images.

### 2.3. Phase Images

The spectral representations of data converted by various transformation techniques have been widely utilized in many signal processing areas to analyze the data with respect to different aspects in a different domain. The Fourier transform is an important method for representing data in the frequency domain and provides two components—magnitude and phase—at every frequency component of the input data. Oppenheim and Lim first presented that more important features of data are preserved in phases of the Fourier transform [26], and Ni and Huo also proved the importance of phase information in signal reconstruction from a statistical point of view [27]. In general, phase-only signals are accentuated where a high temporal (or spatial) frequency change occurs, such as edges or boundaries, without changing their positions in the original data.

In this study, we constructed a phase image from its B-mode image using its own phase information and the representative (or average) magnitude of the Fourier transform, which are averaged for the entire dataset. Because the image sizes of the dataset used in this study are different, the Fourier transform was performed with large enough data points first, and then the reconstructed image was re-sampled to its original size after the

inverse Fourier transform with its own phase and the average magnitude components. The sample phase images calculated from the B-mode image are shown in Figure 2c.

## 3. Methods

In this section, we explain the preprocessing of feature maps and B-mode image for CNN models and the proposed deep learning framework for classification tasks using three time-domain feature maps. The detailed CNN models utilized in our simulation to compare the performances of the traditional CNN and the proposed architecture are explained. The performance metrics used in this study are also summarized.

### 3.1. Preprocessing

Many previous studies on convolutional neural networks have been conducted with a pre-trained CNN followed by fine tuning to obtain classification performance while handling medical images [28]. We utilized several CNN models that were pre-trained from a very large-scale database on the ImageNet dataset that contains over 1.2 million natural images in approximately 1000 classes, and re-trained using the breast US dataset including feature maps for the classification task. Because this work focuses on the classification performances between conventional CNN and the proposed architecture utilizing multiple time-domain feature maps, several variations of VGG, ResNet, and DenseNet, which are popular network models in CNN, were used for comparison.

As explained above, two feature maps, entropy and phase images, were directly calculated from a B-mode image before applying the deep neural network. Because the B-mode images in our dataset are of different sizes, after generating feature maps, all three images were resized to 256 × 256 pixels. There are no specific rules for determination of input image size for the breast US image analysis, but the tradeoff between the diagnostic performance, the training time, and memory needed should be considered. Although our dataset did not provide the detailed dimensions of a scanned image, we selected an image size of 256 pixels in both the axial and lateral directions for visibility of tumors or lesions in several tens of pixels in a B-mode image. To utilize pre-trained CNN models for RGB color images, each image was duplicated, and the input size of a deep neural network was set to 256 × 256 × 3.

### 3.2. Network Architecture

The overall architecture of the traditional and proposed CNN is shown in Figure 3. The traditional CNN for the classification task, shown in Figure 3a, takes B-mode images (256 × 256) as inputs to train the backbone network, and the output vector (1 × 1000) of the backbone network is applied to the linear layer to obtain the final prediction. Various deep neural networks could be used as a backbone network, and the VGG, ResNet, and DenseNet were used in this study to provide reference performance for the BUS classification.

The proposed architecture for BUS classification using multiple time-domain feature maps is shown in Figure 3b. This framework consists of the feature extraction stage and the decision layer along with a backbone network, and we call this architecture a feature-channel convolutional neural network (FC-CNN). The feature extraction stage makes two feature maps, entropy and phase images, from a B-mode image, and adjusts all three images with a size of 256 × 256 pixels as input data for the backbone network. For the backbone networks, three separate CNN models were utilized for each feature-channel image. Each CNN model was independently trained by its own images (i.e., B-mode, entropy, and phase images) and the best model was saved when the highest accuracy occurs. Since entropy and phase images represent distinct characteristics of an image, it would be better to select the most appropriate CNN model for each feature channel respectively for the improvement of performances. However, since the main purpose of this work is to show the benefits of multiple time-domain feature maps derived directly from a B-mode image, the experiments were performed with single CNN model for all feature channels under the same conditions.

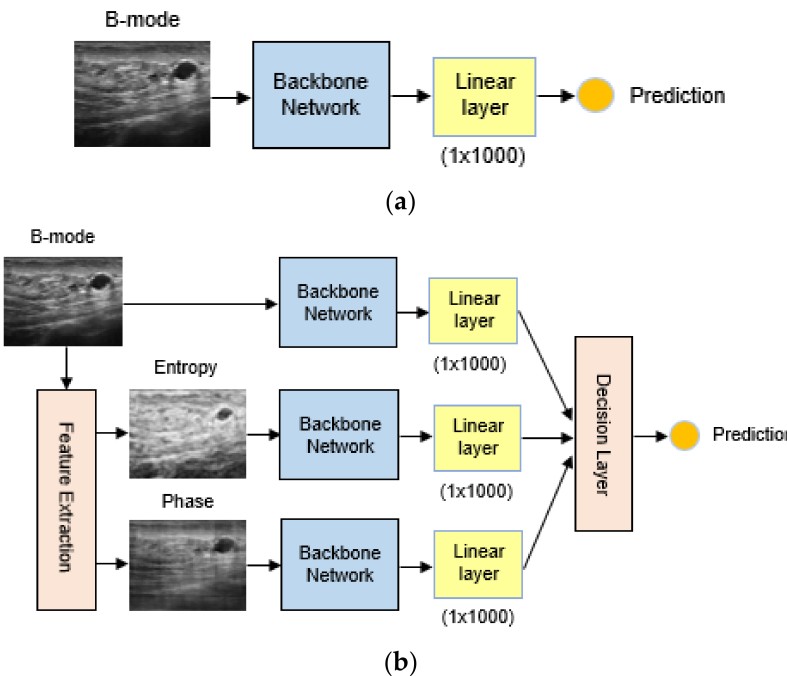

**Figure 3.** Overall architectures of convolutional deep neural networks. (**a**) Traditional convolutional neural network (CNN) using B-mode image only, and (**b**) proposed feature-channel CNN architecture (FC-CNN) using multiple feature images.

The decision layer determines the final prediction for a patient after combining three individual prediction results for B-mode, entropy, and phase images. In this study, to determine a final decision by combining three outputs of each feature-channel, four different combining strategies are utilized that are simple voting, weighted voting, simple averaging, and weighted averaging. Because the output of an individual channel before the activation function consists of three probabilities corresponding to each class, we can use either probabilities or local prediction of each channel for the final classification result. While the simple voting algorithm makes the final prediction by the majority of classes for each channel output, the weighted voting algorithm sets double weight to the channel which has the highest probability in a feature vector. The simple averaging algorithm calculates the mean probability for all three classes first, and then determines the final decision with these mean probabilities. The weighted average algorithm applies the channel accuracy, which was obtained in the training phase, as a weight to the mean probability of each class. The classification performances among combining strategies are compared in the next section.

In this study, several variations of VGG, ResNet, and DenseNet were used as backbone networks to compare the classification performances of the traditional CNN and the proposed FC-CNN. The VGG network is a classical CNN-based deep neural network proposed by the Visual Geometry Group, a research team at Oxford University [29]. For comparison purposes, we utilized relatively simple networks such as VGG-11, VGG-13, and VGG-16, which have fewer layers, because US images contain fewer morphological details than natural images. While the VGG network can extract more detailed information using the smallest filter of $3 \times 3$ and more features of the data by stacking layers deeply, of about 16 and 19 layers, it also experiences the vanishing gradient problem as the layers become deeper [30].

The ResNet proposed by Kaiming He et al. [31] overcomes the vanishing and exploding gradient problems for deeper networks by the initialization or normalization method [32]. The residual block of ResNet solves the accumulated errors in the training stage and improves the nonlinearity by the shortcut connection. In this work, ResNet-18, ResNet-34, and ResNet-50 were simulated as backbone networks. DenseNet is a densely

connected convolutional network that concatenates feature maps in all layers [33]. Because the feature map information is preserved, gradient vanishing problems are alleviated and the training speed is relatively fast owing to the smaller number of parameters. We used DenseNet-121, DenseNet-161, and DenseNet-169 as backbone networks.

### 3.3. Training and Evaluation

To achieve consistency of experiments, we compared all classification performances using a 5-fold cross-validation method, which divides the entire dataset into 80% of the training set and 20% of the evaluation set. Because our dataset consists of an unbalanced number of benign, malignant, and normal cases, the random selection of data for training and evaluation sets from an entire dataset cannot guarantee the fair training of each class. In our simulation, the same distribution ratio of each class was maintained to select the training and evaluation sets, and the classification performances were compared. Since the main purpose of this work is the benefits of multiple time-domain feature maps in CNN-based deep learning, no artificial manipulation was applied to compensate the dataset imbalance and classification performances were compared under the same conditions for the traditional CNN and the proposed FC-CNN methods.

As an augmentation method in the training phase, random numbers from a normal distribution were added to each batch with the same size as the input data, and a few blends of transformations were also applied. The learning process was conducted over 30 epochs when the highest accuracy of validation dataset occurs, and the batch size was set to 8. The CrossEntropyLoss function, which combines the LogSoftmax function and the negative log-likelihood loss function, was used as a loss function, and the Adam optimizer [34] with a learning rate of 1e-4 was also used. Since the proposed feature-channel CNN architecture utilizes three feature maps independently for training each channel backbone network, all weights of the individual CNN model were determined when the highest accuracy was achieved in the training phase. For the experimental environments, PyTorch (v.1.8.0) backend in Python 3.6 was used to implement the networks, and a personal computer outfitted with the Intel Xeon Gold 5120 CPU and NVIDIA Tesla-V100-PCIE graphics processing unit (GPU) was used to conduct all the experiments.

The performance metrics used in this study were accuracy, recall, precision, F1 score, and area under the ROC curve (AUC) [35]. In the following equations, TP, TN, FP, FN, and f(x) represent the number of true positives, true negatives, false positives, false negatives, and the receiver operating characteristic (ROC) curve, respectively.

$$Accracy(\%) = \frac{TP + TN}{TP + TN + FN + FP} \times 100$$

$$Recall(\%) = \frac{TP}{TP + FN} \times 100$$

$$Precision(\%) = \frac{TP}{TP + FP} \times 100$$

$$F1\ score = \frac{2 \times Precision \times Recall}{Precision + Recall}$$

$$AUC = \int_0^1 f(x)dx$$

Multiple feature maps were also effectively utilized in the evaluation stage for classifying BUS images. The B-mode image for evaluation is used to obtain the entropy and phase images with the same parameters, such as the filter size and number of FFT points for the training dataset. Therefore, one patient (or one B-mode image) has three feature images including the original B-mode image, and all the three feature images are applied to the proposed FC-CNN to obtain the classification results separately. As explained above, the final prediction of classification for each evaluation patient (or test B-mode image) was made using the voting algorithm.

## 4. Results

We compared the classification performances of the traditional CNN (with B-mode only) and the proposed FC-CNN (with multiple time-domain feature maps) architecture for nine backbone networks including variations of VGGs, ReNets, and DenseNets. For the performance metrics of accuracy, recall, precision, F1 score, and AUC, all simulation results for classification are shown in Table 2. Note that the highest performance for each metric is represented in bold. Although we used four different combining algorithms in the decision layer to produce the final prediction, the results of the proposed FC-CNN method shown in Table 2 were obtained using the simple voting algorithm, which achieved the best performance. A detailed comparison among the combining strategies is presented in the following section.

**Table 2.** Performances of traditional CNN and the proposed FC-CNN for backbone networks.

| | Backbone | ACC (%) | Recall (%) | Precision (%) | F1 Score (%) | AUC |
|---|---|---|---|---|---|---|
| Traditional CNN | VGG-11 | 89.28 | 87.45 | 87.93 | 87.67 | 0.9427 |
| | VGG-13 | 89.30 | 88.57 | 88.31 | 88.34 | 0.9468 |
| | VGG-16 | 87.97 | 87.43 | 86.28 | 86.62 | 0.9275 |
| | ResNet-18 | 90.07 | 89.95 | 88.56 | 89.07 | 0.9576 |
| | ResNet-34 | 91.50 | 90.73 | **91.29** | 90.85 | 0.9577 |
| | ResNet-50 | 91.24 | 90.64 | 90.99 | 90.79 | 0.9563 |
| | DenseNet-121 | 90.85 | 90.30 | 89.83 | 90.06 | 0.9587 |
| | DenseNet-161 | **92.29** | **92.21** | 91.01 | **91.56** | **0.9685** |
| | DenseNet-169 | 90.72 | 90.85 | 89.28 | 90.02 | 0.9582 |
| Proposed FC-CNN | VGG-11 | 91.76 | 89.34 | 89.70 | 89.38 | 0.9572 |
| | VGG-13 | 91.50 | 89.52 | 89.33 | 89.32 | 0.9562 |
| | VGG-16 | 90.33 | 89.22 | 88.35 | 88.59 | 0.9487 |
| | ResNet-18 | 93.07 | 91.31 | 91.37 | 91.28 | 0.9631 |
| | ResNet-34 | **93.86** | 92.03 | **92.76** | 92.31 | 0.9645 |
| | ResNet-50 | 93.07 | 92.28 | 92.02 | 92.15 | 0.9678 |
| | DenseNet-121 | 92.55 | 91.72 | 91.23 | 91.43 | 0.9645 |
| | DenseNet-161 | 93.73 | **93.19** | 92.40 | **92.76** | **0.9706** |
| | DenseNet-169 | 93.33 | 92.78 | 92.44 | 92.57 | 0.9676 |

While the performance among individual backbone networks made little difference, the proposed FC-CNN generally achieved better performance than the traditional CNN in most performance metrics. As for the accuracy, ResNet backbones showed higher accuracies for the traditional CCN and DenseNet backbones for the FC-CCN architecture, but the differences were not significant. For the performance metrics of recall and precision, we hardly found a trend or tendency according to the backbone networks, but most simulation results of the FC-CNNs were better than those of the traditional CNNs.

To compare the advantages of the proposed FC-CNN architecture in detail, we chose the ResNet-34 network and DenseNet-161 network cases, which achieved the best performance in all aspects. Figure 4 shows the classification accuracies for the traditional CNN with B-mode images and the proposed FC-CNN with multiple time-domain feature maps. The simulation results of the traditional CNN with entropy image and phase image are also plotted for reference. The accuracies for the CNN with a single image (i.e., B-mode, entropy, and phase images separately) are lower than those of the FC-CNN with three feature images simultaneously for both ResNet-34 and DenseNet-161 network models. Since the accuracy of B-mode image is better than those of entropy and phase image for the traditional CNN with a single image, it could be said that B-mode images represent morphological as well as textual information of the tissue scanned. However, the proposed FC-CNN exhibits better performance than the traditional CNN with a B-mode image.

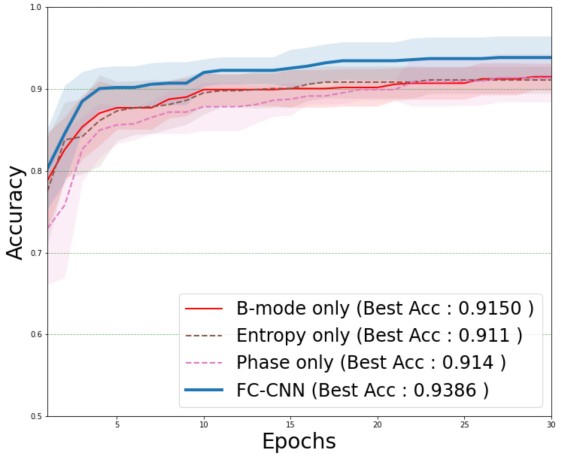 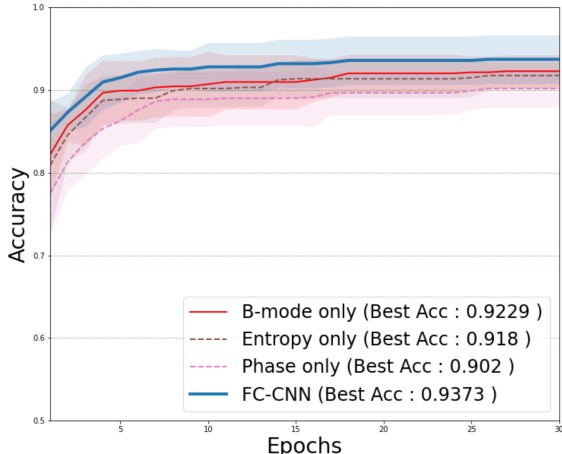

**Figure 4.** The accuracy curves of the traditional CNN using a single image such as B-mode, entropy, and phase images, and the proposed FC-CNN architectures; (**a**) ResNet-34, (**b**) DenseNet-161. Note that the shaded areas represent the five-fold cross-validation results and the solid lines represent their mean accuracies.

The receiver operating characteristic (ROC) curves of the traditional CNN and the FC-CNN architecture for ResNet-34 are shown in Figure 5. Two graphs were plotted for the benign and malignant categories, and entropy and phase-only cases are also shown for reference. As shown in Figure 5, the performances of the traditional CNN with B-mode images and the FC-CNN architecture using multiple feature maps were very similar, but the case of B-mode provided only slightly higher values than the case of FC-CNN for both categories.

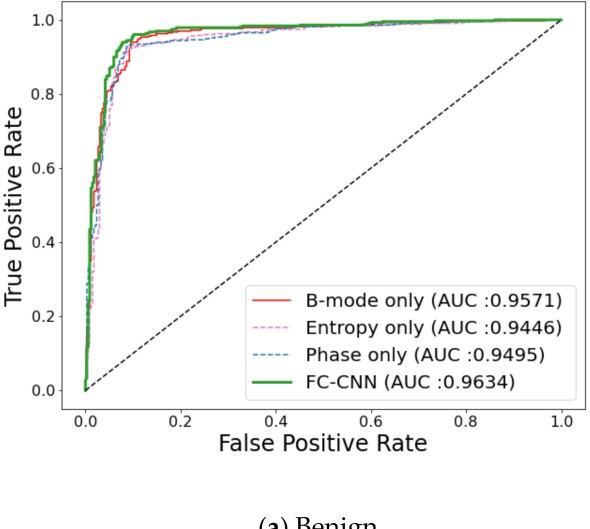 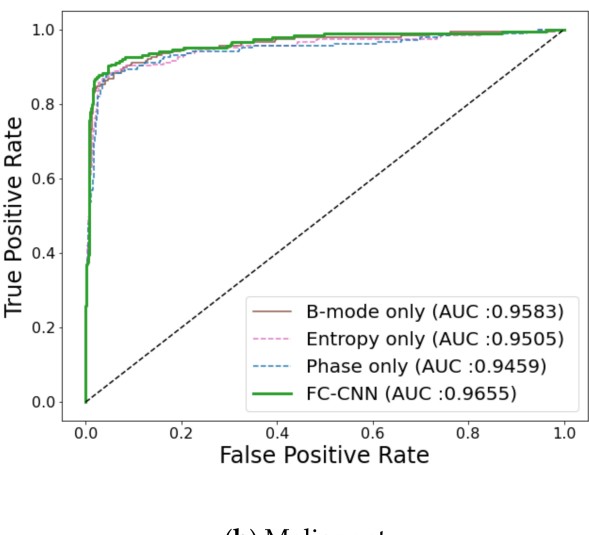

                    (**a**) Benign                                     (**b**) Malignant

**Figure 5.** The ROC Curves of the traditional CNN and the FC-CNN architectures for ResNet-34; (**a**) benign, (**b**) malignant.

For the proposed architecture, three classification results from each feature-channel CNN were properly combined to reduce the classification variances from different feature maps and improve the final prediction result. In this study, we applied four different combining algorithms to individual channel outputs and compared the performances. Table 3 shows the classification accuracy and standard deviation (STD) for the traditional CNN method and the proposed method with four combining strategies which are simple voting, weighted voting, simple averaging, and weighted averaging algorithms. In this

experiment, it is hard to find the significant differences and trends among the combining algorithms, which means all CNN models mostly agree with the same prediction of class.

**Table 3.** Classification accuracy and standard deviation of traditional CNN and FC-CNN combining strategies.

| CNN Model | | Traditional CNN | Simple Voting | Weighted Voting | Simple Averaging | Weighted Averaging |
|---|---|---|---|---|---|---|
| ResNet-34 | Accuracy | 0.915 | 0.939 | 0.929 | 0.935 | 0.928 |
| | STD | 0.016 | 0.026 | 0.023 | 0.026 | 0.016 |
| DenseNet-161 | Accuracy | 0.923 | 0.937 | 0.936 | 0.932 | 0.929 |
| | STD | 0.020 | 0.029 | 0.023 | 0.033 | 0.033 |

## 5. Discussion

The performance of deep learning approaches is generally determined by the architecture of a deep neural network, as well as the quality of the training dataset. In medical US applications, there are few well-annotated datasets available in terms of both varieties (clinical cases, US systems, and institutions) and the amount of data. The classification performance of the proposed method did not exhibit significant difference than the conventional method, because each channel-CNN model was not trained enough for its feature (entropy and phase information) due to small size of training data used. Although entropy and phase images represent distinct features of an image, a sufficient number of training datasets are needed to extract each feature since both images are calculated directly from the same B-mode image. However, the proposed FC-CNN method provided better performances than the traditional method for all CNN models under the same conditions, and the proposed method would be helpful for breast ultrasound classifications when each feature-channel CNN model is well trained with large datasets.

In this study, we proposed one possible way for increasing the number of training data for two feature maps in order to extract (or abstract) its own feature more exactly. Since two feature maps were calculated from a B-mode image, the extended feature maps could be generated using various sizes of filters or different signal processing techniques. For the entropy images, three more sizes of entropy filters which are $9 \times 9$, $15 \times 15$, and $31 \times 31$ pixels, in addition to the entropy filter of $25 \times 25$ pixels, were applied to calculate new entropy images in the training dataset. Since a larger entropy filter relatively smooths local variations of complex structure inside soft tissue, it provides a different view of the tumor area comparing to the background tissue. For the phase images, two additional phase images were reconstructed using one dimensional FFT in either axial or lateral directions with their average magnitudes, and one phase image was obtained using a magnitude of Gaussian distribution. Because the horizontal and vertical edges in a B-mode image are emphasized using axial and lateral 1-D FFT respectively, these phase images also provide additional information of the tissue scanned. Figure 6 shows the examples of extended entropy and phase images. Experimental results for the ResNet-34 model exhibited that classification accuracy with larger extended feature maps outperformed 3.8% higher than that of original feature maps. This shows that final accuracy of the proposed architecture could be improved as the performance of each feature channel improved with larger feature maps.

To compensate the limitations of dataset including the size as well as class imbalance, data augmentation is commonly used to increase the amount of data for the learning process. However, overfitting is a common disadvantage inherited because the data augmentation artificially inflates datasets by image transformations and synthetic creations from a small set of original images. In the proposed architecture, two feature maps, entropy and phase images, along with a B-mode image, represent distinct characteristics that are not easily seen in the original image. Therefore, the proposed method can be considered as

feature-based data augmentation in a broad sense and can be extended for different feature maps with various quantitative ultrasound (QUS) techniques.

Because the entropy and phase images are directly calculated from a B-mode image, the proposed method can be applied to any available ultrasound B-mode dataset to deeply learn the neural networks for various tasks. In addition to feature maps used in this study, other parametric images using the Nakagami distribution, homodyned K-distribution, or any other meaningful models which represent distinct characteristics of tissues and breast masses, would be helpful for the classification of breast ultrasound. For the task of segmentation of tumors in US images, entropy images represent a different aspect of image features compared to a B-mode image and provide additional information to discriminate between the tumor and background tissue. In addition, with more additional feature maps, the deep neural network can classify US images into more detailed categories, such as the American College of Radiology BI-RADS system for breast US images [36].

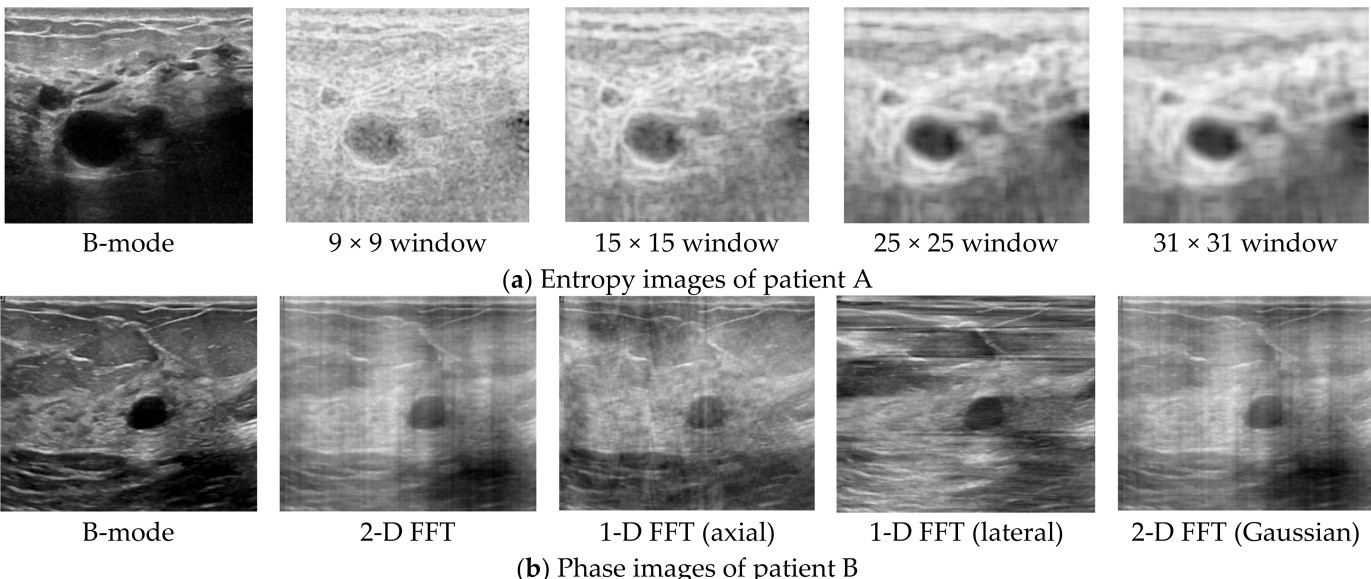

**(a)** Entropy images of patient A

**(b)** Phase images of patient B

**Figure 6.** Examples of extended feature maps using different signal processing techniques; (**a**) entropy images, (**b**) phase images.

## 6. Conclusions

Deep learning-based analysis of medical ultrasound is still lacking compared with the deep learning-based approaches of CT and MRI due to the high variability of operator and/or system dependencies for image acquisition and interpretation. In addition, with a small dataset, a deep neural network would fail to generalize image features for classification. The proposed FC-CNN architecture explicitly extracts distinct features, called feature maps of entropy and phase images, which represent anatomical or microstructural information, obtained from a B-mode image directly, and utilizes them for classification of breast US images to help the deep neural network learn image characteristics quickly and accurately. The experimental results showed that the FC-CNN architecture using multiple time-domain feature maps outperformed the traditional CNN with B-mode only for most of the CNN models. The proposed framework can be extended to other feature maps in both the time and spectral domains and improve the classification performance in deep learning applications.

**Author Contributions:** Conceptualization, H.K. and J.P.; methodology, H.L., G.I., J.L. and K.-B.L.; validation, G.I. and J.P.; formal analysis, H.K. and J.P.; writing—original draft preparation, J.P. and H.K.; writing—review and editing, H.J.L., K.-B.L. and H.K.; supervision, H.J.L. and H.K.; project administration, H.K., K.-B.L. and H.J.L.; funding acquisition, H.J.L. and H.K. All authors have read and agreed to the published version of the manuscript.



**Funding:** This work was supported by the "Human Resources Program in Energy Technology" of the Korea Institute of Energy Technology Evaluation and Planning (KETEP), and granted financial resources from the Ministry of Trade, Industry and Energy, Republic of Korea (No.20194010201830), and a research grant from Kwangwoon University in 2020.

**Institutional Review Board Statement:** Not applicable.

**Informed Consent Statement:** Not applicable.

**Data Availability Statement:** Not applicable.

**Acknowledgments:** This work was supported by the "Human Resources Program in Energy Technology" of the Korea Institute of Energy Technology Evaluation and Planning (KETEP), and granted financial resources from the Ministry of Trade, Industry and Energy, Republic of Korea (No.20194010201830), and a research grant from Kwangwoon University in 2020.

**Conflicts of Interest:** The authors declare no conflict of interest.

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
