# Peer review of "Classification for Breast Ultrasound Using Convolutional Neural Network with Multiple Time-Domain Feature Maps"

_applsci, doi:10.3390/app112110216_

Round 1

Reviewer 1 Report

In this work, the authors propose to use two so-called feature maps (entropy and phase images) derived from B-mode images as inputs to CNN instead of B-mode images only. The authors claim that the proposed deep network model works better with 3-fold information (two features and B-mode images) as inputs than B-mode only models. I have the following reservations.

  1. The B-mode only model performed equally well and there is no significant difference with the proposed feature method. Is the method really helpful to improve the model performance?
  2. Particularly, the authors compare accuracies of ResNet-18 and DenseNet-161 to make their case. However, they have not considered class imbalances in the data set and accuracies may not be the right metric to come to that conclusion.
  3. On what basis, the optimizer, number of epochs were decided? Why any hyperparameter optimization was not done for FC-CNN network?
  4. Advantages of using voting algorithm than any other activation function in the decision layer?

Author Response

The authors appreciate the effort of the reviewer in preparing his/her comments and suggestions during the review of this paper. Please check the attached file for revision.

Reviewer 2 Report

The authors propose the deep learning approach for the classification of ultrasound images, using ensemble learning based convolutional neural network (CNN) architectures.

The authors provide a overview of evolving trends in the field, but the fact is that there are plenty of unmentioned papers that use some variant of CNN ensemble learning to classify ultrasound images, including breast ultrasound images. 

Just some examples, and there are probably even more relevant ones: 
https://iopscience.iop.org/article/10.1088/1361-6560/ab5093/meta
https://www.sciencedirect.com/science/article/abs/pii/S0169260719307059
https://arxiv.org/abs/2102.08567
etc.

It would be correct to place this research in the context of similar approaches. 

As for certain details of the experiment itself, there is quite some ambiguity. For example, how the hyperparameters were determined (there are many of them in the proposed architecture)? Given the minor advances in performance, should the use of some other features be considered? Or significant amounts of augmented images should be used instead of additional features, using other augmentation algorithms?

How is it determined how many epochs the learning phase will last? Is the best model saved? 

A description of the software and hardware used? 

Etc. etc.

Author Response

(The authors gave the same response as above.)

Reviewer 3 Report

In this work, the authors proposed a CNN architecture for breast ultrasound image classification. The novelty is limited, and the manuscript needs an overall revision. The first part of the manuscript is well written, while, starting from section 3, it is less clear and presents some repetitions.

The FC-CNN is not very clear to me, given that it seems that the different types of images are used to train the same backbone network, while I would expect to see a multi-column architecture with three parallel backbones. This point should be addressed in detail by the authors.

The authors mentioned that the dataset is unbalanced, but they have not taken any further steps to address it.

The results reported in section 4 do not present a clear trend; it is unclear if the FC-CNN is beneficial for this task. Indeed the AUC for the classic CNN with DenseNet-161 backbone offer the highest value. I suggest the authors performing other experiments, in particular adding a different dataset.

Author Response

(The authors gave the same response as above.)

Round 2

Reviewer 1 Report

The authors have sufficiently addressed the raised concerns and the revised manuscript can be accepted in the present form.

Author Response

The authors really appreciate all comments and suggestions for the manuscript. We feel that the manuscript has been significantly improved due to reviewer’s efforts. Thank you again for your works in the reviewer of this manuscript.

Reviewer 2 Report

The authors substantially improved their manuscript adding details.

However, at least two dilemmas remain:

1. How to justify the use of the proposed model given the small improvement in performance compared to the base model(s)?

2. The authors concluded: "The classification performance of the proposed method did not exhibit significant difference than the conventional method, because each channel-CNN model was not trained enough for its feature (entropy and phase information) due to small size of training data used.".
How was it concluded that a significantly larger amount of data would result in a greater difference in performance between the proposed model and conventional methods? 

Reviewer 3 Report

The authors revised the manuscript and some details are more clear, but the main point still remains open: is the proposed architecture beneficial for this task? without further experiments, we cannot answer this question.

Round 3

Reviewer 2 Report

Although some dilemmas remain, the authors have made an effort to explain their approach, so the paper is acceptable for publication.

Reviewer 3 Report

The authors revised